# MicroRNA Profiling of Red Blood Cells for Lung Cancer Diagnosis

**DOI:** 10.3390/cancers15225312

**Published:** 2023-11-07

**Authors:** Xinyan Geng, Jie Ma, Pushpa Dhilipkannah, Feng Jiang

**Affiliations:** Departments of Pathology, University of Maryland School of Medicine, Baltimore, MD 21201, USA

**Keywords:** biomarkers, red blood cells, miRNAs, lung cancer, diagnosis

## Abstract

**Simple Summary:**

Despite extensive efforts to identify cell-free circulating biomarkers for lung cancer, none have yet achieved clinical adoption. Recent evidence suggests that blood cell-derived molecules play significant roles in tumorigenesis and may serve as potential diagnostic biomarkers. In this study, we perform microRNA (miRNA) profiling on three primary blood cell types: red blood cells (RBCs), peripheral blood mononuclear cells, and neutrophils, comparing samples from lung cancer patients to healthy controls. We identify distinct miRNA signatures for each cell type and observe significant differences between patient and control cohorts. We show for the first time that RBC-miRNAs have potential as novel biomarkers for lung cancer. By constructing diagnostic panels based on cell-specific miRNAs, we demonstrate that integrating miRNAs from multiple cell sources offers superior diagnostic accuracy than relying on a single cell type.

**Abstract:**

Background: Despite extensive endeavors to establish cell-free circulating biomarkers for lung cancer diagnosis, clinical adoption remains elusive. Noteworthy, emergent evidence suggests the pivotal roles of red blood cells (RBCs) and their derivatives in tumorigenesis, illuminating potential avenues for diagnostic advancements using blood cell-derived microRNAs (miRNAs). Methods: We executed microarray analyses on three principal blood cell types—RBCs, peripheral blood mononuclear cells (PBMCs), and neutrophils—encompassing 26 lung cancer patients and 26 healthy controls. Validation was performed using droplet digital PCR within an additional cohort comprising 42 lung cancer and 39 control cases. Results: Our investigation unearthed distinct miRNA profiles associated with lung cancer across all examined blood cell types. Intriguingly, RBC-miRNAs emerged as potential novel biomarkers for lung cancer, an observation yet to be documented. Importantly, integrating miRNAs from disparate blood cell types yielded a superior diagnostic accuracy for lung cancer over individual cell-type miRNAs. Subsequently, we formulated three diagnostic panels, adeptly discerning non-small cell lung cancer, adenocarcinoma, and squamous cell carcinoma, maintaining consistency across various disease stages. Conclusion: RBC-derived molecules introduce novel cancer biomarkers, and exploiting miRNA profiles across varied blood cell types unveils a promising frontier for lung cancer’s early detection and histological classification.

## 1. Introduction

Analysis of cell-free, tumor-derived molecules in the plasma or serum of blood samples offers a non-invasive approach to the diagnosis, prognosis, and evaluation of therapeutic efficacy in cancer. microRNAs (miRNAs) function as either tumor suppressors or oncogenes whose aberrations are associated with various malignancies. Significant endeavors have been undertaken to identify cancer biomarkers through the detection of circulating extracellular miRNAs originating from tumors [1]. However, challenges persist in the advancement of cell-free biomarkers for cancer. For example, suboptimal miRNA recovery rates, potential contamination from hemolysis, and inconsistencies stemming from the processing and sourcing of cell-free miRNAs can lead to varying or conflicting diagnostic results [2]. The situation is further complicated by the diverse states in which miRNAs are present in the blood, whether bound to proteins like Argonaute complexes and lipoproteins, or enclosed in cellular fragments like exosomes and microvesicles [2,3,4,5,6]. In addition, the cell-free-based biomarkers are limited by the low sensitivity attributed to the scarcity of tumor-derived molecules in the bloodstream, especially during early cancer stages. Moreover, while plasma or serum offers crucial information, it represents only a portion of the intricate matrix that makes up blood, thus constraining the sensitivity of potential biomarkers. Recent studies have shown that various cells within the bloodstream can quickly adapt to stress by modulating the levels of signaling molecules and play pivotal roles in tumorigenesis [7]. For example, leukocytes actively produce and react to a variety of cytokines, which can either amplify or dampen inflammatory processes, crucial events in the oncogenic pathway [8]. Platelets are a rich reservoir of cytokines and growth factors and have emerged as potential diagnostic and therapeutic targets for cancer [9]. We have demonstrated that miRNAs associated with peripheral blood mononuclear cells (PBMCs) and neutrophils possess diagnostic potential for lung cancer [10,11].

Among the different types of blood cells, red blood cells (RBCs), primarily known for their role in respiratory gas exchange, constitute the most abundant cell type in the human body. Despite their prevalence, the role of RBCs in cancer has been surprisingly underexplored [7]. Because of this, along with their high hemoglobin content—which is often viewed as a contaminant in biomarker research—RBCs are considered “junk cells” and discarded in the development of cell-free biomarkers [12,13,14,15]. Recent studies suggest that RBCs, via galectin-4, can directly interact with tumor cells, altering their membrane protein composition and influencing their interactions with platelets throughout carcinogenesis [16,17,18,19,20]. RBCs, using Toll-like receptor 9 on their surface, can attach to cell-free or pathogenic DNA, initiating innate immune responses and inflammation during tumor development and progression [21,22,23,24,25]. Furthermore, changes in RBC indices and oxygen-dependent nitric oxide have been linked to malignancies [26,27,28,29]. In addition, RBCs may harbor DNA fragments and assimilate DNA bearing tumorigenic mutations from cancer cells or tissues through cell-to-cell contact [22]. Moreover, recent studies suggest that RBCs may reach the tumor microenvironment, where intratumor RBCs promote tumor cell proliferation and growth, potentially inducing an immunosuppressive state [30]. It is also proposed that metabolite-induced immunosuppressive functions of RBCs could be initiated within the tumor microenvironment, which plays a crucial role in cancer development and progression [31]. Given that RBCs possess a notably abundant and diverse set of miRNAs compared with many human cells, investigating their role in tumorigenesis could offer pivotal insights into the mechanisms of carcinogenesis, possibly identifying new biomarkers for the disease.

In this study, we simultaneously analyzed miRNA profiles from RBCs, PBMCs, and neutrophils of lung cancer patients and controls. We identified unique miRNA signatures for each cell type, with marked differences between both groups. We showed for the first time that RBC-miRNAs have the potential as novel biomarkers for lung cancer. Using these insights, we developed diagnostic panels combining miRNAs from different cell types, enhancing diagnostic efficacy over using single-cell-type miRNAs.

## 2. Materials and Methods

### 2.1. Patient Cohorts and Specimens

Our research received approval from the Institutional Review Board of the University of Maryland Baltimore. From our pool of participants, we recruited 68 individuals with non-small cell lung cancer (NSCLC), with a breakdown as follows: 17 at stage I, 13 at stage II, 17 at stage III, 18 at stage IV, and 3 cases with unknown stages. Among the NSCLC patients, there were two dominant types: adenocarcinoma (AC) and squamous cell carcinoma (SCC). Additionally, we enrolled 65 smokers who had not been diagnosed with any form of cancer. Our pool of non-cancerous smokers exhibited various lung conditions: granulomatous inflammation (29 individuals), nonspecific inflammatory changes (19 individuals), and lung infections (17 individuals). Aimed at delineating miRNA expression profiles in blood cells, we utilized specimens from 26 individuals diagnosed with NSCLC and an equal number of smokers, forming our exploratory set. Within this subset, we meticulously mitigated potential confounding by matching cases and controls on several pertinent variables, including the number of subjects in each group, sex, race, and smoking history (Table 1). For the validation phase, we used the remaining samples from 42 NSCLC patients and 39 smokers without cancer (Table 1).

### 2.2. Isolation and Processing of Blood Cells

Blood samples were collected into ethylenediaminetetraacetic acid (EDTA) anticoagulant tubes (Becton, Dickinson, Franklin Lakes, NJ, USA) to inhibit clotting and subsequently diluted with phosphate-buffered saline (PBS) (Sigma-Aldrich, St. Louis, MO, USA) at a 1:1 ratio. These diluted samples were delicately layered over Ficoll–Paque in centrifuge tubes, ensuring no mixing occurred with the Ficoll solution. Upon centrifuging at roughly 500× *g* for 30 min at room temperature, distinct layers emerged: the topmost being plasma, followed by PBMCs in the buffy coat, Ficol–Paque in between, and a mix of RBCs and neutrophils at the base. To isolate RBCs, the bottom layer, which consisted of both RBCs and neutrophils, was gently siphoned, with the densest RBCs settling at the very end. The buffy coat was directly pipetted to collect PBMCs into a fresh tube. To isolate neutrophils, the remaining section of the bottom layer (post-RBC extraction) was combined with 3 parts of cold PBS and then centrifuged at 300× *g* for 10 min at 4 °C. The resulting pellet was momentarily resuspended in a hypotonic solution to remove any lingering RBCs, followed by the infusion of a 1.8% saline solution. After another centrifugation, the now neutrophil-rich pellet was reconstituted in PBS.

### 2.3. Assessment of Purity of Isolated RBCs, PBMCs, and Neutrophils

We used magnetic-activated cell sorting (MACS) technology to evaluate the purity of the isolated RBCs, PBMCs, and neutrophils. RBCs were resuspended in MACS buffer (Miltenyi Biotec. San Ramon, CA, USA) and incubated with magnetic beads coated with antibodies against glycophorin A, a distinctive RBC marker, as per the manufacturer’s guidelines for MACS kits (Miltenyi Biotec). For PBMC validation, the isolated PBMCs were similarly resuspended in MACS buffer and treated with beads conjugated to antibodies specific for CD3 (T cells) and CD19 (B cells). In the case of neutrophils, they were resuspended in MACS buffer and exposed to beads coated with CD66b antibodies, a specific neutrophil marker. The purity of each isolated cell type was determined using flow cytometry (Becton, Dickinson), as described in previous research [32]. Controls, including cells without labels (assessing non-specific binding) and cells tagged with unrelated antibodies (acting as negative controls), were consistently integrated into the experiments.

### 2.4. RNA Isolation

RNA was isolated from cell samples using methodologies detailed in our prior works [10,11]. We evaluated the purity and concentration of the isolated RNA using OD260/280 readings on a dual-beam UV spectrophotometer (Eppendorf AG, Hamburg, Germany), aiming for an optimal ratio between 1.8 and 2.0. The integrity of the RNA samples was assessed using capillary electrophoresis with the RNA 6000 Nano Lab-on-a-Chip kit and a Bioanalyzer 2100 (Agilent Technologies, Santa Clara, CA, USA). Only RNA samples with integrity numbers exceeding 7 were considered for subsequent analyses.

### 2.5. Real-Time PCR-Based Microarray Analysis of miRNAs

The examination of miRNA expression profiles within blood cells was performed by Exiqon Services (Exiqon, Denmark), following the methodologies as previously outlined [10,11]. Briefly, 6 μL of RNA underwent reverse transcription in 30 μL reactions, using the miRCURY LNA™ Universal reverse transcription (RT) miRNA PCR, Polyadenylation, and cDNA synthesis kit (Exiqon). The synthesized cDNA was then diluted 100-fold and assayed in 10 μL PCR reactions, consistent with the miRCURY LNA™ Universal RT miRNA PCR protocol. Each miRNA underwent a singular qPCR assay on the miRNA Ready-to-Use PCR, Human Panel I, utilizing the ExiLENT SYBR^®^ Green master mix. Control reactions, devoid of the template during the reverse transcription, were executed and profiled akin to the samples. The amplifications were undertaken within a LightCycler^®^ 480 Real-Time PCR System (Roche, San Francisco, CA, USA) in 384-well plates. Amplification curves were derived using the quantification cycle (Cq) values, which subsequently served as relative quantifiers for the assayed genes. To normalize the data, we used the average of assays detected in the samples, calculated as (average–assay Cq). We then used the statistical methods delineated in the subsequent statistical analysis section to identify genes differentially expressed between NSCLC patients and cancer-free smokers.

### 2.6. Droplet Digital PCR (ddPCR) Analysis of miRNAs

We conducted ddPCR evaluations of the miRNAs, adopting the protocols as previously described [33,34,35,36]. In brief, from each sample, 1 μL of RNA was subjected to RT using the TaqMan miRNA RT Kit (Applied Biosystems, Foster City, CA, USA) and gene-specific primers, producing the corresponding cDNA. The specific primers used for the ddPCR evaluation of miRNAs are cited in our previous studies [37,38,39]. A 20 μL reaction mixture, composed of 5 μL cDNA solution, 10 μL Supermix, and 1 μL of the Taqman primer/probe mix, was introduced into a cartridge containing Droplet Generation Oil (Bio-Rad, Hercules, CA, USA). Subsequently, the cartridge was positioned within the QX100 Droplet Generator (Bio-Rad). The resulting droplets were methodically transferred to a 96-well PCR plate and subjected to PCR amplification using the T100 thermal cycler (Bio-Rad). Leveraging the count of positive reactions and Poisson’s distribution, a direct and reliable quantification of the original target concentration was ascertained.

### 2.7. Statistical Analysis

To detect differentially expressed miRNAs between lung cancer patients and cancer-free controls in the exploratory set, we used the following parameters to determine sample size: a permissible false positive rate of 1.0, an anticipated fold difference of 2.0 between normal and tumor samples, a standard deviation of 0.7 for gene measurements on a base-two logarithmic scale, and an intended power of 80%. With the inclusion of 372 miRNAs in the array, a total of 26 specimens per group were deemed necessary to meet the stipulated statistical criteria. Subsequent to this, to validate and develop miRNA marker panels for distinguishing between cancer patients and controls, we used the Receiver Operator Characteristic (ROC) curve with an emphasis on the Area Under the Curve (AUC) for sample size determination. With an established AUC of H0 at 0.5 (representing our null hypothesis), and under the alternative hypothesis of H1, a minimum of 28 subjects in each category was requisite to discern a noteworthy difference, aiming for an AUC of 0.75 against an AUC of 0.5, ensuring 80% power at a 5% significance level. Consequently, our sample composed of 42 NSCLC patients juxtaposed with 39 cancer-free controls suffices to provide robust statistical power for biomarker development. Microarray data were analyzed using the Kruskal–Wallis test or the Mann–Whitney U test. A False Discovery Rate (FDR)-adjusted *p*-value of less than 0.05 was considered significant. Further refinement of miRNA analysis was undertaken using the Least Absolute Shrinkage and Selection Operator (LASSO) method, aiming to pinpoint the most efficacious combination of miRNAs for diagnostic biomarker panels. The selected miRNA panel was subjected to Pearson’s correlation analysis, ensuring the mitigation of collinearity. The diagnostic efficacy of the chosen miRNA sets underwent rigorous examination using multivariate binary logistic regression, paired with ROC and AUC evaluations, thereby elucidating predictive algorithms, sensitivities, and specificities. Relationships between miRNAs and variables such as age, sex, race, smoking pack years, and cancer stages were determined using multivariate analysis of variance.

## 3. Results

### 3.1. Isolation and Purification of Blood Cells and RNA Samples

From the 135 blood samples collected, we effectively isolated three distinct cell types: RBCs, PBMCs, and neutrophils. Flow cytometry analysis confirmed that the purity of these isolated cell types exceeded 95%. Moreover, the RNA extracted from these cell populations was of high purity, evidenced by a 260/280 ratio ranging between 1.8 and 2.0. The quality of the RNA was also superior, with an RNA integrity number consistently ≥ 7.

### 3.2. Distinct miRNA Patterns Can Differentiate among Various Blood Cell Types

Among the 372 mature human miRNAs present on the Exiqon qPCR-based miRNA array, an average of 212 miRNAs were detected per sample across the three different cell populations, as they exhibited a Cq value of <37. Fifty of the miRNAs exhibited significant differential expression (an FDR-adjusted *p*-value of less than 0.05) across the three cell populations from cancer-free smokers (Figure 1 and Appendix A).

### 3.3. Distinct Cell-Specific miRNA Profiles Associated with Lung Cancer Patients

In RBCs of NSCLC patients, a differential expression was observed for six miRNAs (FDR-adjusted *p*-values < 0.05) compared with cancer-free smokers (Figure 2A and Table 2). Particularly, miR-93-5p and miR-449b-5p were expressed at elevated levels, whereas miRs-29c-3p, 15a-5p, 449a, and 148a-3p showed reduced levels relative to controls. In PBMCs, three miRNAs exhibited varied expression levels (all *p* < 0.05) in NSCLC patients as compared with cancer-free smokers (Figure 2B and Table 2). Of the miRNAs, miRs-576-3p and 19b-3p were found at higher levels, while miR-29b-3p showed a decreased expression in NSCLC patients (*p* < 0.05). In neutrophils, differential expression was noted for three miRNAs in NSCLC patients in comparison with cancer-free smokers (all *p* < 0.05), as detailed in Figure 2C and Table 3. Specifically, miRs-423-3p and 574-3p were at heightened levels, and miR-26a-2-3p was present at a diminished level relative to the controls.

### 3.4. Distinct miRNA Profiles in Cell Types for Differentiating Lung Cancer from Controls

To validate our miRNA array findings, we utilized ddPCR, a different technique, on a separate cohort comprising 42 lung cancer patients and 39 controls. Each of the 12 miRNAs produced at least 10,000 droplets in each sample well, ensuring an effective ddPCR “reading” for absolute gene quantification. This established that all 12 miRNAs could be consistently measured using the ddPCR technique.

From these 12 miRNAs, 9 exhibited changes in line with the miRNA array data (Table 3). Among the eight ncRNAs, PBMC miR-19b-3p displayed an association with the age of the subjects (*p* = 0.045) (Appendix A). RBC miR-15a-5p, PBMC miR-19b-3p, neutrophil miR-26a-2-3, and neutrophil miR-574-3p showed a significant association with patient gender (all *p* < 0.05) (Appendix A). RBC miR-29c-3p, PBMC miR-19b-3p, and PBMC miR-29b-3p were related to race (*p* = 0.005). Additionally, PBMC miR-29b-3p and neutrophil miR-26a-2-3p were associated with the histology of lung cancer (*p* < 0.05) (Appendix A). Nonetheless, these miRNAs did not exhibit any relationships with other patient attributes, smoking history and size of pulmonary nude, or tumor stages (Appendix A).

We used logistic regression and a backward elimination approach to identify specific miRNA biomarkers of the cell types for lung cancer. In RBCs, three miRNAs (miRs-93-5p, 29C-3p, and 449-5p) were identified, which, used in combination, produced an AUC of 0.76 for the diagnosis of lung cancer, being significantly higher than that of any single one of the miRNAs (Figure 3A, Table 4 and Appendix A) (*p* < 0.05). As a result, the analysis of the three RBC miRNAs produced 77.42% sensitivity and 68.29% specificity for the diagnosis of NSCLC. Furthermore, four miRNAs, including miRs-15a-5p, 93-5p, 29C-3p, and 449-5p, were identified as a panel of biomarkers that diagnosed AC with 0.84 AUC, producing a sensitivity of 87.50% and specificity of 80.49% for the diagnosis of AC (Figure 3B, Table 4 and Appendix A). In addition, miRs-93-5p, 29c-3p, and 15a-5p, and 449b-5p in RBCs were identified as a panel of biomarkers for SCC with 0.7363 AUC, 71.43% sensitivity, and 80.49% specificity (Figure 3C, Table 4 and Appendix A).

In PBMCs, three miRNAs (miR-576-3p, 19b-3p, and 29b-3p) used in combination yielded an AUC of 0.7452, which was significantly higher than that of any single one of the miRNAs (Figure 3D, Table 4 and Appendix A) (*p* < 0.05). As a result, this panel of biomarkers could diagnose lung cancer with a sensitivity of 70.59% and a specificity of 65.79%. Furthermore, miR-29b-3p used alone in PBMCs could detect AC with 0.7266 AUC, producing a sensitivity of 42.86% and a specificity of 67.44% (Figure 3E and Table 4). In addition, the three PBMC miRNAs (miRs-576-3p, 19b-3p, and 29b-3p) were developed as a panel of PBMC miRNA biomarkers with an AUC of 0.85 in distinguishing SCC cases from controls (Figure 3F and Table 4). Subsequently, the panel of the three PBMC miRNA biomarkers had 72.73% sensitivity and 86.49% specificity for SCC (Table 4).

In neutrophils, when used in combination, two miRNAs (miR-26a-2-3p and miR-574-3p) displayed an AUC of 0.6919 (Figure 3G and Table 4) (*p* < 0.05). Consequently, the combined use of the two miRNAs generated 64.10% sensitivity and 63.64% specificity for the detection of NSCLC (Table 4). Furthermore, using neutrophil miR-26a-2-3p alone had an AUC of 0.75 in distinguishing AC cases from controls with 53.33% sensitivity and 66.67% specificity (Figure 3H and Table 4). However, no miRNA tested in neutrophils could efficiently distinguish SCC cases from controls with an AUC.

### 3.5. Integrated miRNAs from Distinct Blood Cell Types Exhibit Synergistic Effects on Lung Cancer Detection

We used logistic regression models with a constraint approach to determine whether combining miRNAs from distinct blood cell types enhances lung cancer diagnosis. An integrated panel of biomarkers including two RBC miRNAs (miRs-93-5p and 29c-3p), two PBMC miRNAs (miRs-576-3p and 19b-3p), and two neutrophil miRNAs (miRs-26a-2-3p and 574-3p) produced a greater AUC (0.87) than did the individual cell type biomarker (*p* < 0.05) (Figure 3I and Table 4). The integrated panel of biomarkers had a sensitivity of 82.86% and a specificity of 78.38% for the diagnosis of lung cancer (Table 4).

Furthermore, four RBC miRNAs (miRs-93-5p, 29c-3p, and 15a-5p, and 449b-5p) and one neutrophil miRNA (miR-26a-2-3p) were identified as an integrated panel that had an AUC of 0.90 for the diagnosis of AC with 85.00% sensitivity and 86.49% specificity (Figure 3J and Table 4). In addition, three RBC miRNAs (miRs-29c-3p, 15a-5p, and 449b-5p), three PBMC miRNAs (miRs-576-3p, 19b-3p, and 29b-3p), and one neutrophil miRNA (miR-26a-2-3p) used in combination produced 0.9495 AUC with 81.82% sensitivity and 89.19% specificity for the detection of SCC (Figure 3K and Table 4). These integrated panels of biomarkers had no association with the stage of lung cancer, and patients’ age, race, gender, smoking history, and size of pulmonary nodule (all *p* > 0.05).

## 4. Discussion

Lung cancer remains the foremost cause of cancer-related deaths, with NSCLC constituting 85% of all lung cancer incidences, predominantly composed of AC and SCC. Early detection can decrease mortality [40]. Substantial endeavors have been undertaken to advance cell-free plasma miRNAs as diagnostic biomarkers for lung cancer. This includes research spearheaded by Zaporozhchenko et al., as well as our own investigations [34,39,41]. However, none have been adopted in clinical settings due to challenges such as suboptimal miRNA recovery rates from plasma or serum and variability in cell-free miRNA sources. Blood cell-derived miRNAs, with their rich and stable source, might sidestep these obstacles. It is posited that blood cells could contribute to tumorigenesis through both direct and indirect pathways. In this present study, we delineated the miRNA profiles of RBCs from both cancer patients and controls, revealing that RBC-centric miRNAs could serve as novel biomarkers for lung cancer, which was previously unreported. Simultaneously, we profiled the miRNAs of PBMCs and neutrophils, contrasting them with RBCs, and identified unique blood cell-specific miRNAs linked to lung cancer. Harnessing this knowledge, we crafted cell-type-specific miRNA panels as promising biomarkers for diagnosing lung cancer. Moreover, a holistic approach—integrating miRNA data from various blood cells—outperformed individual biomarker panels in diagnosing NSCLC. Our tailored diagnostic panels also proved highly effective for detecting both AC and SCC. Importantly, these biomarker panels worked well across different stages of NSCLC and for various sizes of lung nodules. These biomarkers hold the potential for early-stage detection of NSCLC and offer improved classification of specific types.

These blood cell miRNAs identified as potential biomarkers may have pivotal roles in lung tumorigenesis. For instance, miR-93-5p is linked to enhanced cell proliferation, migration, and a poor prognosis in lung cancer [42]. The miR-449 family impacts NSCLC development by targeting IL-6 and affecting the JAK2/STAT3 signaling pathway [43]. The miR-29 family advances lung cancer progression through the AMPK/mTOR signaling pathway by negatively regulating MTFR1 [44], while miR-15a-5p inhibits metastasis and lipid metabolism in NSCLC [45]. Interestingly, miR-148a-3p, found to be significantly down-regulated in lung tissues, can hinder lung adenocarcinoma progression by targeting MAP3K9 [46]. Other miRNAs, such as miR-576-3p, miR-19b-3p, and miR-29b, are associated with migration inhibition, tumor progression facilitation, and modulation of lung cancer progression, respectively [46,47,48]. Moreover, miR-423-3p and miR-574-5p are tied to tumor growth promotion in lung adenocarcinoma [49,50]. Lastly, miR-26a-2-3p augments the metastatic capability of lung cancer cells via the AKT pathway [51]. Especially given that RBCs might play intricate and crucial roles in cancer development, it is vital to consistently investigate how these miRNAs affect tumorigenesis within the framework of RBC regulation.

In this study, we adhered to the National Cancer Institute’s Early Detection Research Network biomarker guidelines [52], which emphasize the importance of confirming and validating miRNAs detected with array-based platforms using alternative methods that are more reliable and cost-effective. We used ddPCR for the validation of miRNAs initially identified with the array-based platform. We selected ddPCR due to its enhanced reliability and cost-effectiveness, making it a commonly utilized technique for analyzing genetic changes associated with cancer. Notably, ddPCR eliminates the need for internal control genes, as required by array techniques, and additionally enables precise quantification of miRNA copy numbers per microliter. Although a deeper understanding of miRNA stability and normalization processes remains essential, this methodology may offer the potential to address the challenges linked to array techniques and present a cost-effective avenue for early cancer detection.

Intriguingly, the present study reveals significant associations among specific miRNAs and factors such as age, gender, race, and histology in the context of lung cancer (Appendix A). Particularly, PBMC miR-19b-3p’s association with subject age suggests its potential value in understanding age-related variations in lung cancer. Elevated miRNA expression in different age groups might lead to tailored diagnostic and therapeutic approaches. Additionally, RBC miR-15a-5p, PBMC miR-19b-3p, neutrophil miR-26a-2-3, and neutrophil miR-574-3p, displaying gender-related associations, raise intriguing possibilities for gender-specific considerations in lung cancer. Tailored screening strategies for male and female subjects may enhance early detection. Moreover, the associations among RBC miR-29c-3p, PBMC miR-19b-3p, and PBMC miR-29b-3p with race underscore the importance of recognizing potential racial disparities in lung cancer and miRNA profiles. This highlights the need for diversity-sensitive diagnostic and therapeutic strategies. Furthermore, associations between PBMC miR-29b-3p and neutrophil miR-26a-2-3p and lung cancer histology suggest distinct miRNA expression patterns among different subtypes, potentially guiding subtype-specific diagnostic and treatment approaches. However, validation using further studies is essential to confirm the clinical utility of these miRNAs as biomarkers. Additionally, elucidating the underlying biological mechanisms will be critical for effective translation into clinical practice.

While our findings are promising, our study has several limitations. First, our chosen array specifically targets only 372 miRNAs to determine differentially expressed miRNAs in the blood cells of NSCLC patients, potentially overlooking key lung cancer-associated miRNAs that are not present on the platform or not reliably measurable with our method. Utilizing a more comprehensive tool, such as whole-genome next-generation sequencing, may unveil new cell-type-specific miRNA signatures for NSCLC, thus refining the miRNA-based assay for lung cancer. Second, this study’s sample size is modest, and the samples are sourced from clinically diagnosed hospital patients, not fully representing heavy smokers typically screened for early lung cancer detection. It underscores the need to validate these biomarkers in a comprehensive, multisite lung cancer screening trial. Third, although miRNA profiles across various blood cell types offer promise as innovative diagnostic tools for early lung cancer detection, it is imperative to concurrently compare cell-based miRNAs with cell-free plasma miRNAs. This comprehensive approach is essential for the development of a unified diagnostic strategy for lung cancer. Furthermore, while our study was conducted with a validated cohort size to ensure the robustness of the biomarkers, it is also necessary to conduct a thorough examination of the factors influencing miRNA variation in blood cells, explore potential sources of intrinsic variation, and provide detailed insights into the methods used to control for confounding variables.

## 5. Conclusions

In summary, our research uncovered distinctive miRNA profiles across various blood cell types, notably in RBCs, revealing significant disparities between cancer patients and control groups. Furthermore, we developed diagnostic panels integrating cell-specific miRNAs, which potentially offer superior diagnostic precision over individual cell-type miRNAs. However, thorough subsequent studies are imperative to authenticate these initial findings.

## Figures and Tables

**Figure 1 cancers-15-05312-f001:**
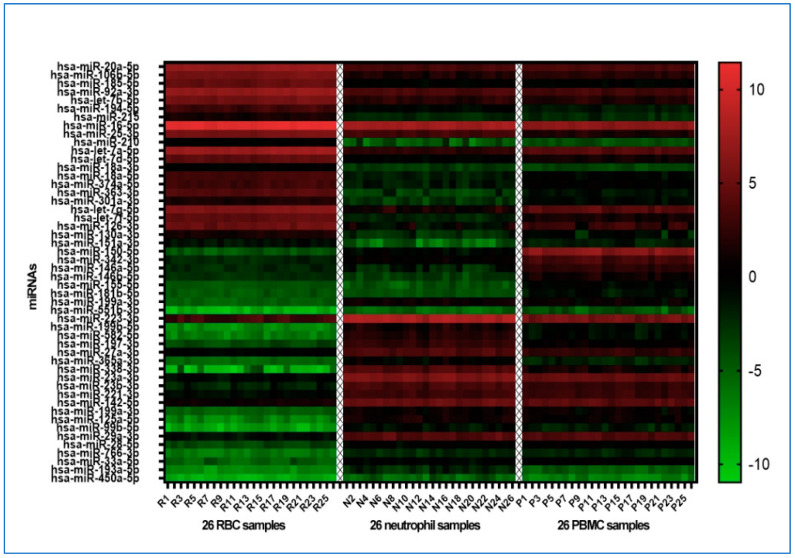
A heatmap showcasing the differential expression of the 50 miRNAs with the most significant standard deviation across three distinct cell populations from the cancer-free smokers. The color gradient moves from green (denoting down-regulation) to black (indicating no change) and culminates in red (highlighting up-regulation). The results of normalized dCq values were analyzed using the Kruskal–Wallis test. A False Discovery Rate (FDR)-adjusted *p*-value of less than 0.05 was considered significant. Each row corresponds to a specific miRNA, while each column signifies a unique sample.

**Figure 2 cancers-15-05312-f002:**
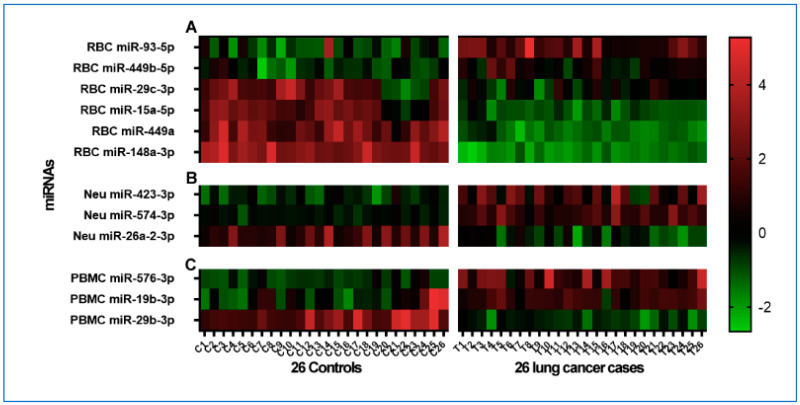
Heatmaps displaying unique miRNA profiles for each cell population, comparing lung cancer patients to cancer-free controls. The color gradient transitions from green (down-regulation) to black (no change) and culminates in red (up-regulation). Expression data rely on normalized dCq values. The Mann–Whitney U test was used for statistical analysis, with significance set at an FDR-adjusted *p*-value of <0.05. Within the heatmap, each row signifies a unique miRNA, while columns denote individual samples. Labels (**A**–**C**) indicate miRNA changes in RBCs, neutrophils (Neu), and PBMCs, respectively, across the 26 cancer-free smokers and 26 lung cancer patients.

**Figure 3 cancers-15-05312-f003:**
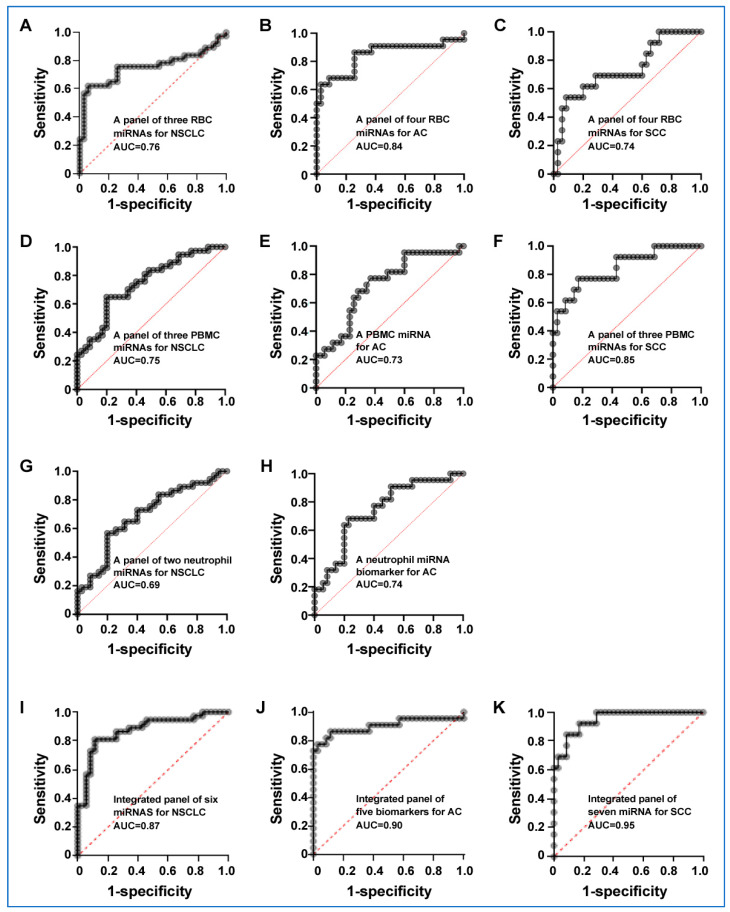
Receiver Operating Characteristic (ROC) curve analysis for panels of miRNAs in differentiating NSCLC patients from cancer-free smokers. The ROC curve showcases the diagnostic performance of these biomarker panels, with the Area Under the Curve (AUC) indicating their overall diagnostic accuracy. (**A**) A panel of RBC miRNA biomarkers for detecting NSCLC. (**B**) A panel of RBC miRNA biomarkers for diagnosing AC. (**C**) A panel of RBC miRNA biomarkers for diagnosing SCC. (**D**) A panel of PBMC miRNA biomarkers for diagnosing NSCLC. (**E**) A PBMC miRNA biomarker (miR-29b-3p) for diagnosing AC. (**F**) A panel of PBMC miRNA biomarkers for diagnosing SCC. (**G**) A panel of neutrophil biomarkers for diagnosing NSCLC. (**H**) A neutrophil miR-26a-2-3p for diagnosing AC. (**I**) Integrated panel of biomarkers for diagnosing NSCLC. (**J**) Integrated panel of biomarkers for diagnosing AC. (**K**) Integrated panel of biomarkers for diagnosing SCC.

**Table 1 cancers-15-05312-t001:** Characteristics of NSCLC patients and cancer-free smokers in the exploratory set and validation set.

An Exploratory Set			A Validation Set		
	NSCLC cases (*n* = 26)	Controls (*n* = 26)		NSCLC cases (*n* = 42)	Controls (*n* = 39)
Age	66.73 (SD 11.25)	65.782 (SD 10.39)	Age	65.38 (SD 10.23)	64.74 (SD 10.26)
Sex			Sex		
Female	9	9	Female	12	11
Male	17	17	Male	30	28
Race			Race		
AAs	11	11	AAs	26	25
WAs	16	16	WAs	16	14
Smoking pack-years (median)	32.7	30.9	Smoking pack-years (median)	33.8	31.2
Pulmonary nodule size (mm)	21.26 (SD 10.25)	6.72 (SD 3.58)	Pulmonary nodule size (mm)	20.25 (SD 11.38)	6.63 (SD 3.97)
Stage			Stage		
Stage I	7		Stage I	10	
Stage II	5		Stage II	8	
Stage III	8		Stage III	9	
Stage IV	6		Stage IV	12	
			Unknown	3	
Histological type			Histological type		
Adenocarcinoma	15		AC	26	
SCC	11		SCC	16	

Abbreviations: NSCLC, non-small cell lung cancer; AC, adenocarcinoma; SCC, squamous cell carcinoma; SD, standard deviation.

**Table 2 cancers-15-05312-t002:** The 12 miRNAs found to be significantly differential expressed in three cell populations of lung cancer patients with cancer-free controls. The results were analyzed using the Mann–Whitney U test with an FDR-adjusted *p*-value of less than 0.05.

miRNAs	Mean Expression (Controls)	Mean Expression (Cancer Patients)	Mann–Whitney U Statistic	FDR-Adjusted *p*-Value
RBC miR-93-5p	0.922	1.529	41	<0.001
RBC miR-449b-5p	−0.585	0.370	116	<0.001
RBC miR-29c-3p	−0.585	−0.145	189	0.006 *
RBC miR-15a-5p	2.039	−1.041	10	<0.001 *
RBC miR-449a	2.431	−1.299	0	<0.001 *
RBC miR-148a-3p	2.728	−1.701	0	<0.001 *
Neutrophil miR-423-3p	−0.464	1.550	75	<0.001
Neutrophil miR-574-3p	−0.255	1.014	2	<0.001
Neutrophil miR-26a-2-3p	0.926	−0.454	6	<0.001 *
PBMC miR-576-3p	−0.754	1.476	16	<0.001
PBMC miR-19b-3p	−0.105	1.219	126	<0.001
PBMC miR-29b-3p	1.739	−0.528	0	<0.001 *

*, the miRNA was significantly decreased in expression in lung cancer patients compared with cancer-free smokers.

**Table 3 cancers-15-05312-t003:** The nine observed miRNA changes detected using ddPCR in an additional 42 cases and 39 controls, consistent with the miRNA array data. The results were analyzed using the Mann–Whitney U test with an FDR-adjusted *p*-value of less than 0.05.

miRNAs	Mean Expression (Controls)	Mean Expression (Cancer Patients)	Mann–Whitney U Statistic	FDR-Adjusted *p*-Value
RBC miR-93-5p	38.443	52.000	474	0.009
RBC miR-449b-5p	0.016	0.022	631	0.008
RBC miR-29c-3p	3.930	3.030	561	0.014 *
RBC miR-15a-5p	151.015	112.129	506	0.003 *
PBMC miR-576-3p	0.045	0.056	576	0.021
PBMC miR-19b-3p	24.834	27.016	515	0.004
PBMC miR-29b-3p	8.220	6.855	438	0.001 *
Neutrophil miR-574-3p	0.605	0.790	532	0.010
Neutrophil miR-26a-2-3p	0.031	0.019 *	397	0.003 *

*, the miRNA was significantly decreased in expression in lung cancer patients compared with cancer-free smokers.

**Table 4 cancers-15-05312-t004:** The diagnostic values of the individual panels and the integration for lung cancer.

	AUC, % (95% CI)	Sensitivity, % (95% CI)	Specificity, % (95% CI)
A panel of three RBC miRNA biomarkers for NSCLC	0.76 (0.64 to 0.87)	77.42% (58.90% to 90.41%)	68.29% (51.91% to 81.92%)
A panel of four RBC miRNA biomarkers for AC	0.84 (0.72 to 0.96)	87.50% (61.65% to 98.45%)	80.49% (65.13% to 91.18%)
A panel of four RBC miRNA biomarkers for SCC	0.74 (0.57 to 0.90)	71.43% (29.04% to 96.33%)	80.49% (65.13% to 91.18%)
A panel of three PBMC miRNA biomarkers for NSCLC	0.75 (0.63 to 0.86)	70.59% (52.52% to 84.90%)	65.79% (48.65% to 80.37%)
A PBMC miRNA biomarker for AC	0.73 (0.59 to 0.86)	57.14% (28.86% to 82.34%)	67.44% (51.46% to 80.92%)
A panel of three PBMC miRNA biomarkers for SCC	0.85 (0.72to 0.97)	72.73% (39.03% to 93.98%)	86.49% (71.23% to 95.46%)
A panel of two neutrophil miRNA biomarkers for NSCLC	0.69 (0.57 to 0.82)	64.10% (47.18% to 78.80%)	63.64% (45.12% to 79.60%)
A neutrophil miRNA biomarker for AC	0.74 (0.61 to 0.87)	53.33% (26.59% to 78.73%)	66.67% (50.45% to 80.43%)
Integrated panel of biomarkers for NSCLC	0.87 (0.79to 0.96)	8056%, (63.98% to 91.81%)	83.33% (67.19% to 93.63%
Integrated panel of biomarkers for AC	0.90 (0.79 to 1.00)	85.00% (62.11% to 96.79%)	86.49% (71.23% to 95.46%)
Integrated panel of biomarkers for SCC	0.95 (0.89 to 1.00)	81.82% (48.22% to 97.72%)	89.19% (74.58% to 96.97%)

## Data Availability

The data that support the findings of this study are available from the corresponding author upon reasonable request.

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
