# Peer review of "MicroRNA Profiling of Red Blood Cells for Lung Cancer Diagnosis"

_cancers, 2023, doi:10.3390/cancers15225312_

Round 1

Reviewer 1 Report

Comments and Suggestions for Authors

This article by Geng et al. focuses on cell-based biomarkers instead of serum/plasma biomarkers, a not so studied field. Therefore, I believe it is quite novel and original (especially since the authors took into account erythrocytes as well) and the findings can feed future, larger studies for the emergence of novel biomarkers. Moreover, it perfectly fits the scopes of the journal.

I only have some minor comments:

Introduction Section: The authors discuss miRNAs as biomarkers and also paint a picture of the role of erythrocytes in the neoplasmic context. I believe that the Introduction is quite complete, but I would suggest the addition of a recent perspective (10.2174/1573394718666220428120818) that focuses on the role of intratumor erythrocytes in immunosuppression. The authors could enrich their paragraph regarding red blood cells by adding some information from this article.

Results Section: Please enlarge Figure 3 and the font used, because it's quite hard to read as it is.

Discussion Section: The discussion section is on-point and I really like the fact that the authors give information regarding the miRNAs that stood out. Could the authors discuss a little bit more the findings that are presented in Supplemental Table 2? The associations between the miRNAs and age, sex etc. are really interesting. Could they be useful in the clinical setting? For example, could some miRNAs be more sensitive in the diagnosis of female subjects or young subjects?

Author Response

I only have some minor comments:

Introduction Section: The authors discuss miRNAs as biomarkers and also paint a picture of the role of erythrocytes in the neoplastic context. I believe that the Introduction is quite complete, but I would suggest the addition of a recent perspective (10.2174/1573394718666220428120818) that focuses on the role of intratumor erythrocytes in immunosuppression. The authors could enrich their paragraph regarding red blood cells by adding some information from this article.

Responses: In response to your valuable suggestion, we have incorporated the recent perspective (Charalampos et al.,) which delves into the role of intratumor erythrocytes in immunosuppression, into the Section of Introduction. This addition would strengthen our discussion of red blood cells in the context of neoplasms.

Results Section: Please enlarge Figure 3 and the font used, because it's quite hard to read as it is.

Responses: Yes, we have enlarged Figure 3 and adjusted the font size as recommended.

Discussion Section: The discussion section is on-point and I really like the fact that the authors give information regarding the miRNAs that stood out. Could the authors discuss a little bit more the findings that are presented in Supplemental Table 2? The associations between the miRNAs and age, sex etc. are really interesting. Could they be useful in the clinical setting? For example, could some miRNAs be more sensitive in the diagnosis of female subjects or young subjects?

Responses: We are grateful for your insightful feedback on our discussion section. Acting on your suggestions, we have enhanced our discourse regarding the implications of miRNAs. We've further discussed the potential clinical repercussions of miRNA associations with demographic variables, specifically age and gender. Emphasis has been placed on their possible roles as markers in distinct demographic groups, such as females and younger populations. These revelations could pave the way for personalized medicine strategies in the clinical realm. Nonetheless, it is imperative to undertake further validation and prospective studies to substantiate the clinical applicability of these miRNAs as biomarkers. Additionally, a comprehensive understanding of the biological mechanisms driving these associations is paramount for their effective integration into clinical practice.

Reviewer 2 Report

Comments and Suggestions for Authors

The work presented in this study by Geng et al. “MicroRNA Profiling of Red Blood Cells for Lung Cancer Diagnosis” is aimed at identifying a panel of potential novel biomarkers for lung cancer based on cell specific-miRNAs. They show promising results and, in particular, that combining miRNA profiles across various red blood cell types offers superior diagnostic accuracy than relying on a single cell type.

-       This study was well designed and validation of the miRNA candidates as biomarkers was confirmed by two independent techniques and in two different cohorts. In addition, a robust statistical analysis underscores the results obtained. Although interesting, this study extends and refines with slight variations what few groups reported and published like Zaporozhchenko et al, 2018 for instance who profiled 179 miRNA expression in blood plasma of lung cancer patients as in in non-cancer individual and identified a lung tumour-specific subsets of miRNAs.

-       A general question that still remains to be better argued or discussed in this Ms is to which extent, beyond correlation analysis only, variation in the expression of miRNA in blood cells reflects a phenomenom really linked to cancer specific origin rather than intrinsic independent variation in blood cells. Although authors achieved their study by validating that the size of the cohort was enough.

-        

-       For Real-time PCR-based microarray analysis of miRNAs authors used the average of assays detected in the samples. Fifty miRNAs out of 212 detected (372 in the array) exhibited significant differential expression among which 12 were found as good biomarkers. However, 9 were confirmed using the ddPCR analysis. Although panel 1 contains miRNAs that are more expressed, why authors did not use panel 2 to assess whether they would obtain close to 100% overlap with the two techniques which would have strengthened even more their analysis rather than discussing about the benefit of a whole-genome next-generation sequencing to assign a cell-type-specific miRNA signatures for NSCLC. Indeed, the more expressed miRNAs can thus be considered also as the more stable ones, having a stronger impact on the mRNA they might deregulate compared to less stable ones, but normalization might lead to a bias when comparing cancer and control populations, particularly if they have a very different average.  Authors should present the data as supp.info or at least have to comment on that specific point.

Author Response

-       This study was well designed and validation of the miRNA candidates as biomarkers was confirmed by two independent techniques and in two different cohorts. In addition, a robust statistical analysis underscores the results obtained. Although interesting, this study extends and refines with slight variations what few groups reported and published like Zaporozhchenko et al, 2018 for instance who profiled 179 miRNA expression in blood plasma of lung cancer patients as in in non-cancer individual and identified a lung tumor-specific subsets of miRNAs.

Response:  

We appreciate the suggestion to compare our work with that of Zaporozhchenko et al. However, it is essential to underscore that our research is not merely an extension or refinement of previous works. As outlined in the Background and Discussion Sections of our manuscript, there have been significant strides in the quest to pinpoint cell-free plasma miRNA biomarkers, encompassing studies by Zaporozhchenko et al., as well as our own endeavors. Yet, the journey to harness cell-free plasma cancer biomarkers presents notable challenges. These include issues related to suboptimal miRNA retrieval, variability stemming from processing and sourcing, the intricate states of miRNAs in circulation, and the limited sensitivity of cell-free biomarkers, particularly during the early phases of cancer. All these factors require meticulous attention if we are to maximize the potential of such biomarkers.

In contrast, blood cell-derived miRNAs offer a more stable and reliable source, potentially overcoming these challenges. Our present study concurrently analyzed miRNA profiles from RBCs, PBMCs, and neutrophils in lung cancer patients and controls, leading to the identification of unique miRNA signatures for each cell type. Notably, we provided evidence for RBC-miRNAs as novel lung cancer biomarkers and developed diagnostic panels that combine miRNAs from different cell types to enhance diagnostic efficacy. Thus, our research stands distinct from those centered on profiling miRNA expression in the plasma of lung cancer patients. As a result, we believe our approach may pave the way to overcoming the obstacles in creating effective cell-free plasma biomarkers for cancer diagnosis.

In response to the reviewer's concern, in the Discussion Section of the revised manuscript, we acknowledge the importance of comparing cell-based miRNAs with cell-free plasma miRNAs to develop a comprehensive diagnostic strategy for lung cancer.

-       A general question that still remains to be better argued or discussed in this Ms is to which extent, beyond correlation analysis only, variation in the expression of miRNA in blood cells reflects a phenomenon really linked to cancer specific origin rather than intrinsic independent variation in blood cells. Although authors achieved their study by validating that the size of the cohort was enough.

Response: Thank you for highlighting this crucial aspect. In our continued research, we are delving into a meticulous examination of the factors contributing to miRNA variation in blood cells. We will also investigate the potential origins of intrinsic variation and provide a detailed account of the methodologies employed to account for confounding factors. The revised manuscript now incorporates this discussion, both in response to your feedback and to elevate the overall caliber of our study.

-       For Real-time PCR-based microarray analysis of miRNAs authors used the average of assays detected in the samples. Fifty miRNAs out of 212 detected (372 in the array) exhibited significant differential expression among which 12 were found as good biomarkers. However, 9 were confirmed using the ddPCR analysis. Although panel 1 contains miRNAs that are more expressed, why authors did not use panel 2 to assess whether they would obtain close to 100% overlap with the two techniques which would have strengthened even more their analysis rather than discussing about the benefit of a whole-genome next-generation sequencing to assign a cell-type-specific miRNA signatures for NSCLC. Indeed, the more expressed miRNAs can thus be considered also as the more stable ones, having a stronger impact on the mRNA they might deregulate compared to less stable ones, but normalization might lead to a bias when comparing cancer and control populations, particularly if they have a very different average.  Authors should present the data as supp.info or at least have to comment on that specific point.

Response: In our research, we diligently adhered to the protocols set forth by the National Cancer Institute's Early Detection Research Network (NCI-EDRN), as detailed by Srivastava et al. These directives underscore the paramount significance of not only identifying but also rigorously validating miRNAs pinpointed via array-based platforms. Such validation is best achieved using alternative methodologies that are more dependable and cost-efficient. To fulfill this imperative, we turned to digital droplet polymerase chain reaction (ddPCR) as our validation tool, post the preliminary identifications made through the array-based platform. Our choice of ddPCR was influenced by its established credibility, cost-effectiveness, and its prevalent use in discerning genetic alterations associated with cancer. Notably, ddPCR overcomes a significant constraint of array-based techniques: it obviates the need for internal control genes. Furthermore, ddPCR offers a granular quantification, measuring miRNA copy numbers per microliter, thereby effectively countering the shortcomings inherent to array-based approaches.

As mentioned in our paper, we have plans to leverage whole-genome next-generation sequencing to assign cell-type-specific miRNA signatures for non-small cell lung cancer. Furthermore, your suggestion regarding the utilization of panel 2 to assess overlap between different techniques is well-founded. In our ongoing research, we intend to explore the application of panel 2 to further validate our findings and evaluate the concordance between microarray and ddPCR techniques.

Additionally, we acknowledge the significance of considering miRNA stability and its potential impact on mRNA regulation. In our ongoing project, we are committed to addressing this specific aspect, exploring the implications of miRNA stability, and the normalization process within our analysis. To provide clarity and transparency, we have explicitly acknowledged these points in the revised manuscript, accompanied by detailed responses within the Discussion Section.

Reviewer 3 Report

Comments and Suggestions for Authors

If you analyze and compare test results for blood cells other than RBC, you may be able to draw a clearer conclusion.

Comments on the Quality of English Language

If you analyze and compare test results for blood cells other than RBC, you may be able to draw a clearer conclusion.

Author Response

If you analyze and compare test results for blood cells other than RBC, you may be able to draw a clearer conclusion.

Response: In our present study, we have indeed explored and analyzed data from various blood cell types beyond RBCs. This includes the examination of miRNA profiles peripheral blood mononuclear cells (PBMCs) and neutrophils. Our analysis has revealed distinct miRNA signatures for each cell type, and we have observed significant differences between the patient and control cohorts. Notably, our study demonstrates for the first time that the integration of miRNAs from multiple cell sources results in superior diagnostic accuracy compared to relying solely on a single cell type.